# Novel Relation Detection: Discovering Unknown Relation Types via Multi-Strategy Self-Supervised Learning

**Qingbin Liu**[1]*, **Yin Kung**[1]*, **Yanchao Hao**[1], **Dianbo Sui**[2], **Siyuan Cheng**[3], **Xi Chen**[1]†,
**Ningyu Zhang**[3], **Jiaoyan Chen**[4]

[1] Platform and Content Group, Tencent, China
[2] Harbin Institute of Technology, Weihai, China
[3] Zhejiang University & AZFT Joint Lab for Knowledge Engine, Zhejiang, China
[4] Department of Computer Science, The University of Manchester, UK
{qingbinliu, amykung, marshao}@tencent.com, suidianbo@hit.edu.cn,
sycheng@zju.edu.cn, jasonxchen@tencent.com,
zhangningyu@zju.edu.cn, jiaoyan.chen@manchester.ac.uk

## Abstract

Conventional approaches to relation extraction can only recognize predefined relation types. In the real world, new or out-of-scope relation types may keep challenging the deployed models. In this paper, we formalize such a challenging problem as Novel Relation Detection (NRD), which aims to discover potential new relation types based on training samples of known relations. To this end, we construct two NRD datasets and exhaustively investigate a variety of out-of-scope detection methods. We further propose an effective NRD method that utilizes multi-strategy self-supervised learning to handle the problem of shallow semantic similarity in the NRD task. Experimental results demonstrate the effectiveness of our method, which significantly outperforms previous state-of-the-art methods on both datasets.

## 1 Introduction

Relation extraction (RE) is an important task in structured information extraction, which aims to recognize the relations of entity pairs from texts (Riedel et al., 2013; Zeng et al., 2014; Lin et al., 2016). For example, given the sentence "*Westworld is a science fiction western series directed by Jonathan Nolan.*" and the entity pair [*Jonathan Nolan*, *Westworld*], an RE model should output the relation type "*the director of*".

Existing RE methods typically follow the closed-world classification assumption and can only recognize predefined relation types. However, such an assumption limits the usage of these methods in real-world applications, as new or out-of-scope (OOS) relation types may continually emerge after the model is deployed. For example, in the Wikidata knowledge graph (Vrandecic and Krötzsch,

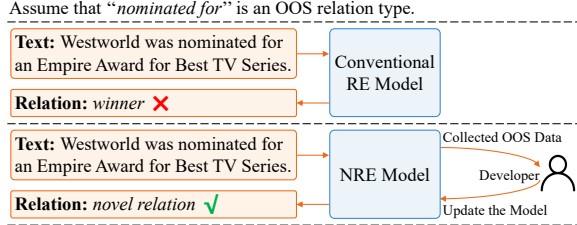

Assume that "*nominated for*" is an OOS relation type.

**Text:** Westworld was nominated for an Empire Award for Best TV Series.

**Relation:** winner ✗

Conventional RE Model

**Text:** Westworld was nominated for an Empire Award for Best TV Series.

**Relation:** novel relation ✓

NRE Model

Collected OOS Data

Developer

Update the Model

Figure 1: An example of novel relation extraction. For the data of the OOS relation, the conventional RE model predicts a wrong relation. The NRE model is able to correctly predict the novel relation type.

2014), new items and properties keep appearing over time[1]. These new relations may mislead the deployed RE model, making it incorrectly assign known relations to the data of new relations, as shown in Figure 1. In addition, existing methods cannot automatically discover new relations for future development.

To handle this problem, we propose a more realistic and challenging task, Novel Relation Extraction (NRE), which aims to discover potential new relation types based on training samples of known relations. Note that we define novel relations as OOS relations that are not included in the predefined relation set. In the NRE task, we group the novel relation types into a new class and convert the traditional $k$-class RE task into a $(k + 1)$-class RE task. The $(k + 1)$-th class represents the novel relation. This task requires RE models to correctly identify not only the known relations but also the novel relation. Based on the OOS data collected by the NRE model, developers can easily and purposefully annotate the data and update the model, as shown in Figure 1.

The novel relation class in the NRE task is differ-

---

*Equal Contribution.
†Corresponding author.

[1] https://www.wikidata.org/wiki/Wikidata:News

ent from the "no relation" class and the "other relation" class in existing RE tasks (Hendrickx et al., 2010; Zhang et al., 2017). Previous developers always assume that the "no relation" class and the "other relation" class are known in the development phase, and they can annotate a large amount of training data for these two relations to train RE models. However, the novel relation class indicates that the text contains new or OOS relations whose distribution is unknown and unpredictable in the development phase. It is infeasible for developers to annotate training data for the novel relation class in real-world applications since real test data is usually unknown and continually changing.

There are two lines of previous work related to NRE, open RE (Yao et al., 2011; Marcheggiani and Titov, 2016; Wu et al., 2019; Liu et al., 2021) and OOS detection (Hodge and Austin, 2004; Chandola et al., 2009; Lin and Xu, 2019; Wu et al., 2021). Open RE means that there are no predefined relation types and no labeled data. Open RE extracts phrases and arguments as specific relations and discovers new relations by clustering or heuristics. Compared to open RE, NRE strengthens the capability of the conventional RE methods and aims to automatically discover novel relation types that are not in the predefined relation set, while providing accurate predictions for known relations. Another line of related work is OOS detection, which aims to recognize OOS data that does not belong to any predefined class. Although OOS detection has been widely investigated in other NLP tasks, its exploration in RE is relatively rare.

Therefore, in this paper, we formalize the NRE task and construct two NRE datasets based on two widely used RE datasets, FewRel (Han et al., 2018) and TACRED (Zhang et al., 2017). Then, to establish NRE's baselines, we exhaustively investigate a variety of OOS detection methods (Hendrycks and Gimpel, 2017; Lin and Xu, 2019; Yan et al., 2020). In general, previous OOS detection methods usually learn the decision boundaries of known classes based on the feature or probability distributions of known training data. In the testing phase, they use confidence scores to identify samples outside the decision boundaries as OOS data.

However, when applying existing OOS detection methods in the NRE task, we find a shallow semantic similarity problem. Specifically, sentences with OOS relations may have similar surface information, such as entity overlapping and similar syntac-

Assume that "*nominated for*" is an OOS relation type.

| |
|---|
| **Text 1 (OOS):** Westworld was nominated for an Empire Award for Best TV Series. |
| **Text 2 (Known):** Westworld is directed by Jonathan Nolan. |

Figure 2: An example of shallow semantic similarity. Although "Text 1" and "Text 2" express different relations, there is similar surface information between them.

tic structures, to sentences with known relations, as shown in Figure 2. Previous methods only use training samples of known relations to train models, which makes them difficult to handle OOS data with similar surface information. They may predict similar features or probabilities for these OOS data and known training data, eventually leading to confusion between the novel and known relations.

To address the above problem, we propose an effective NRE method, called Multi-Strategy Self-Supervised Learning (M3S), which explicitly models OOS data with similar semantic information by constructing pseudo-OOS data. Specifically, M3S uses features of known training data to construct various pseudo-OOS data. In the semantic feature space, these pseudo data are close to the known data. Then, M3S utilizes these pseudo-OOS data to train the model, thus improving the generalization in the novel relation. Experimental results show that M3S significantly outperforms previous state-of-the-art OOS detection methods.

In summary, the contributions of this work are as follows: (1) To the best of our knowledge, we are the first to formally introduce OOS detection into RE and we construct two NRE benchmarks through two widely used RE datasets. (2) We investigate a variety of existing OOS detection methods and further propose multi-strategy self-supervised learning, which can effectively handle the problem of shallow semantic similarity in the NRE task. (3) Experimental results show that M3S significantly outperforms previous OOS detection methods on the two benchmarks. The source code and benchmarks will be released for further research (https://github.com/liuqingbin2022/NRE).

## 2 Task Formulation

### 2.1 Relation Extraction

The traditional RE task is usually formulated as a text classification task (Zhang et al., 2017; Han et al., 2018). Given a sentence $x$ that contains a pair of entities, the traditional RE task is to predict a

relation $y$ for these two entities. This task assumes $y \in \mathcal{Y}$, where $\mathcal{Y}$ denotes a predefined relation set.

## 2.2 Novel Relation Extraction

The NRE task aims to identify the data of OOS relations while correctly classifying the data of known relations. We denote the data of known relations as $\mathcal{D}^{\text{o}} = (\mathcal{D}_1^{\text{o}}, \mathcal{D}_2^{\text{o}}, ..., \mathcal{D}_k^{\text{o}})$. $\mathcal{D}_i^{\text{o}}$ is the data of the $i$-th known relation, which has its own training, validation, and test sets ($\mathcal{D}_i^{\text{train,o}}$, $\mathcal{D}_i^{\text{valid,o}}$, $\mathcal{D}_i^{\text{test,o}}$). To match the realistic environment, there is no training and validation data for the novel relation type. In the development phase, the NRE model can only access the training and validation data of known relations. In the test phase, the NRE model should classify each sample into $(k + 1)$-class relations, where the $(k + 1)$-th class is the novel relation type. Therefore, in the test phase, we employ the test data of both known and novel relations to evaluate the model. Due to the lack of OOS training data, it is difficult for previous RE methods to make accurate predictions for the novel relation type.

## 3 Dataset

To the best of our knowledge, we are the first to formally introduce the NRE task. Therefore, we construct two NRE datasets based on two widely used RE datasets. In this section, we first briefly introduce the original RE datasets and then describe the construction method of the NRE datasets. Finally, we show the statistics of these datasets.

## 3.1 Original Relation Extraction Datasets

FewRel (Han et al., 2018) is a few-shot RE dataset, which contains 100 relations. In our work, we use the publicly available 80 relations as the original dataset. Since FewRel contains a sufficient number of relation types, we can use FewRel to simulate various OOS relations well. Each relation in the FewRel dataset has 700 labeled data. TACRED (Zhang et al., 2017) is an RE dataset that contains 42 relations. In the TACRED dataset, there are 68124, 22631, and 15509 samples in the training, validation, and test sets, respectively.

## 3.2 NRE Dataset Construction

For an original RE dataset, we randomly select some relations as OOS relations according to a specific ratio. Since many relations can be predefined in practical applications, we adopt three reasonable ratios, 10%, 20%, and 30%, in this paper. For

| Metric | FewRel-NRE-20% | TACRED-NRE-20% |
|---|---|---|
| # Training Data | 26,880 | 2,880 |
| # Validation Data | 8,960 | 960 |
| # Test Data | 11,200 | 1,200 |
| # OOS Test Data | 2,240 | 240 |
| Vocabulary Size | 66,171 | 13,714 |
| # Known Relations | 64 | 16 |
| # OOS Relations | 16 | 4 |

Table 1: Statistics of the NRE datasets. "#" indicates "the number of".

these OOS relations, we remove their training and validation data and keep their test data as the test data of the novel relation type. Based on FewRel and TACRED, we propose two instantiations of the above construction method. FewRel-NRE: Since the original FewRel dataset does not provide data splitting, we split the data of each relation in a 3:1:1 ratio into the training, validation, and test sets. TACRED-NRE: Considering the severe class imbalance in TACRED, we use the top 20 most frequent relations and limit the number of training, validation, and test samples of each relation to 180, 60, and 60, respectively. We treat the "no relation" class in TACRED as a known relation to fit the real-world setting. To avoid randomness, we construct 5 different datasets with 5 different random seeds for each specific ratio. These datasets can well simulate unpredictable and diverse OOS relations in the real world. Table 1 shows the statistics of the two constructed datasets in which 20% of the relations are selected as OOS relations.

## 4 Previous OOS Detection Methods

In other OOS detection tasks, recent work has attempted to find an appropriate decision boundary to balance the performance of both known and OOS relations. We roughly divide previous methods into two categories: probability-based methods (Hendrycks and Gimpel, 2017; Shu et al., 2017) and feature-based methods (Lin and Xu, 2019; Yan et al., 2020). We apply these methods to the NRE task and provide a detailed quantitative analysis.

## 4.1 Probability-Based Methods

Probability-based methods assume that the updated model will not be overconfident in the OOS data. In the training phase, probability-based methods use the training data of known relations to update the model. In the test phase, these methods derive a confidence score from the probability distribution of each sample. If the confidence score of a test

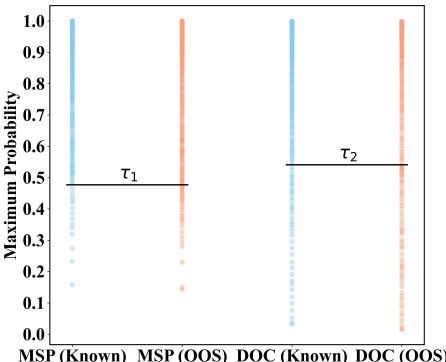

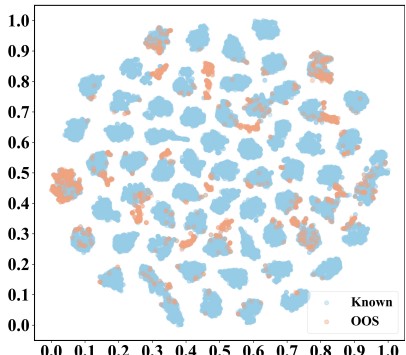

Figure 3: Maximum probabilities of the test samples of the FewRel-NRE-20% dataset. "MSP (Known)" denotes the maximum probabilities predicted by the MSP model for test samples with known relations. Test samples whose maximum probability is below the threshold $\tau$ are predicted as OOS data.

Figure 4: t-SNE visualization of the FewRel-NRE-20% dataset. Blue dots denote the features of the training samples with known relations. Light red dots denote the features of the OOS test samples.

sample is below a threshold, it is treated as OOS data. For example, MSP (Hendrycks and Gimpel, 2017) adopts the maximum softmax probability as the confidence score.

However, we find that these methods perform poorly in the NRE task. To give a clear analysis, we present the maximum probabilities of test samples with known relations and OOS relations separately in Figure 3. From the results, we can see that both the MSP and DOC (Shu et al., 2017) methods tend to overconfidently assign known relations to OOS data. We speculate that since these models have not learned the OOS data, they may not generalize well to the OOS data with similar surface information, leading to incorrect predictions. Besides, these methods require an additional validation set containing OOS data to obtain the threshold ($\tau$).

### 4.2 Feature-Based Methods

Feature-based methods derive the confidence score from the feature distribution. In the training phase, these methods employ specific optimization objectives to constrain the feature distribution of known classes. In the test phase, they use distance-based outlier detection algorithms, such as LOF (Breunig et al., 2000), to derive the confidence score from the feature distribution. For example, SEG (Yan et al., 2020) is a typical feature-based method that assumes the features of known classes follow a Gaussian mixture distribution.

To verify the effectiveness of feature-based methods on the NRE task, we provide the feature visualization of the SEG method in Figure 4. We can see that many features of OOS relations are con-

fused with features of known relations. Since the SEG method has not learned OOS data, it may have difficulty extracting valid features for these OOS samples, which ultimately hurts the performance.

## 5 Methodology

To address the above problem, we propose Multi-Strategy Self-Supervised Learning, which explicitly models OOS data with similar surface information by constructing various pseudo-OOS data.

### 5.1 Input Encoding

We adopt BERT (Devlin et al., 2019), which is a powerful pre-trained language model, as the text encoder. The BERT encoder outputs a contextual representation for each input sequence as:

$$\boldsymbol{h}_x = \text{BERT}(x) \tag{1}$$

where $\boldsymbol{h}_x$ is the hidden state. We use the hidden state of the [CLS] token as the feature representation ($f(x)$) of each sequence. Note that our method is agnostic to the model architecture. Other encoders can also be adopted.

### 5.2 Multi-Strategy Self-Supervised Learning

In M3S, we design three self-supervised strategies to construct pseudo-OOS data, including convex combination, regional replacement, and irregular replacement. These pseudo-OOS data, together with the training data of known relations, are used to update the model to improve its generalization. The framework of M3S is shown in Figure 5.

### 5.2.1 Convex Combination

For this strategy, we use convex combinations of sample features from different relations as pseudo-

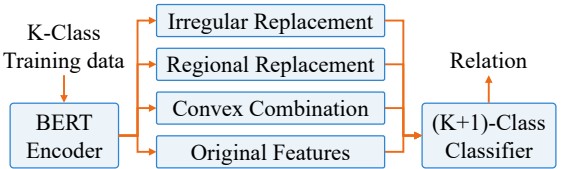

Figure 5: Overall framework of the proposed method.

OOS data, which can be formalized as:

$$f^{\text{cc}} = \alpha * f(x_1) + (1 - \alpha) * f(x_2) \qquad (2)$$

where $x_1$ and $x_2$ are two training samples with different known relations. These two samples are randomly selected from the same training batch. $f^{\text{cc}}$ is the synthetic OOS data. $\alpha$ is a scalar that is randomly sampled from a uniform distribution $\text{U}(0, 1)$. If $\alpha$ is close to 0 or 1, $f^{\text{cc}}$ will be close to the features of known relations in the semantic feature space. These synthetic features can thus simulate real OOS data that have similar semantic information as known training data.

These synthetic features are assigned to the novel relation type to form a new training set: $\mathcal{D}^{\text{train,cc}} = \bigcup_{i=1}^{N} f_i^{\text{cc}}$. Since these samples are constructed in the feature space, it is very efficient to construct a large number of pseudo data.

### 5.2.2 Regional Replacement

Since the distribution of real OOS data is often arbitrary and diverse, we further propose a regional replacement strategy to construct diverse pseudo-OOS data. This strategy constructs pseudo samples by replacing a random region of the feature of $x_1$ with the corresponding region of the feature of $x_2$:

$$f^{\text{rr}} = \boldsymbol{M} \odot f(x_1) + (\boldsymbol{1} - \boldsymbol{M}) \odot f(x_2) \qquad (3)$$

where $\boldsymbol{M}$ is a binary mask vector that indicates the region of replacement. $x_1$ and $x_2$ are two training samples with different known relations. $\odot$ is the element-wise multiplication.

To get the mask vector $\boldsymbol{M}$, we first sample a random value $\beta$ from a uniform distribution $\text{U}(0, 1)$. Then, we calculate the region length as: $l = \beta * d$, where $d$ is the dimensional size of features. Finally, we calculate the region coordinates:

$$\begin{aligned} t &\sim \text{U}(1, d) \\ r^{\text{s}} &= t - \lfloor l/2 \rfloor, \ r^{\text{e}} = t + \lfloor l/2 \rfloor \end{aligned} \qquad (4)$$

where $t$ is a random integer between 1 and $d$. $r^{\text{s}}$ and $r^{\text{e}}$ are the start and end coordinates of the selected region. In this region, the value of the mask vector is 0, otherwise it is 1. This strategy forms another new OOS training set: $\mathcal{D}^{\text{train,rr}} = \bigcup_{i=1}^{N} f_i^{\text{rr}}$.

### 5.2.3 Irregular Replacement

Moreover, we design an irregular replacement. Different from regional replacement, irregular replacement randomly selects some discontinuous dimensions to replace, instead of a continuous region.

To obtain the mask vector $\boldsymbol{M}$ of this strategy, we first sample the random value $\beta$ and calculate the number of replaced dimensions as: $l = \beta * d$. Then, we assign a coefficient $\gamma$, which is randomly selected from the uniform distribution $\text{U}(0, 1)$, for each dimension. Finally, we select the top $l$ dimensions that get larger coefficients for replacement. In the mask vector, we set these selected dimensions to 0 and the others to 1. We use the mask vector to construct the pseudo-OOS data $f^{\text{ir}}$ and obtain a new training set: $\mathcal{D}^{\text{train,ir}} = \bigcup_{i=1}^{N} f_i^{\text{ir}}$.

### 5.3 Optimization

In each training batch, we combine the original features with the synthetic OOS features as the training set, i.e., $\mathcal{D}^{\text{train}} = \mathcal{D}^{\text{train,o}} \cup \mathcal{D}^{\text{train,cc}} \cup \mathcal{D}^{\text{train,rr}} \cup \mathcal{D}^{\text{train,ir}}$. Thus, we can train a uniform $(k + 1)$-class classifier without additional post-processing. In our method, we employ the softmax classifier and use the cross-entropy loss to optimize the model:

$$\mathcal{L}_{\text{CE}} = -\sum_{i=1}^{|\mathcal{D}^{\text{train}}|} \log \frac{\text{e}^{\phi_j(f_i)/T}}{\sum_{n=1}^{|\mathcal{C}|} \text{e}^{\phi_n(f_i)/T}} \qquad (5)$$

where $\phi_j(f_i)$ denotes the output logit of the ground-truth class $j$ of the feature $f_i$. $\mathcal{C}$ is the entire $(k + 1)$-class relation set. $T$ is the temperature scalar. In the testing phase, M3S is able to directly predict known relations while identifying the novel relation.

## 6 Experiments

### 6.1 Baselines

To provide a comprehensive comparison, we employ multiple OOS detection methods as baselines.

**MSP** (Hendrycks and Gimpel, 2017) obtains the confidence score from the maximum softmax probability and treats samples that get lower scores as OOS data. **DOC** (Shu et al., 2017) uses multiple 1-vs-rest sigmoid classifiers to optimize the probability distribution. **LMCL** (Lin and Xu, 2019) utilizes a large margin cosine loss to learn discriminative features and detects outliers via the LOF algorithm. **SEG** (Yan et al., 2020) assumes that the features of known classes follow a Gaussian mixture distribution. It also uses LOF to detect outliers. **SEG-SF** (Yan et al., 2020) uses a softmax

| Rate | Method | FewRel | | | | TACRED | | | |
|---|---|---|---|---|---|---|---|---|---|
| | | Accuracy | Macro F1 | Known | Novel | Accuracy | Macro F1 | Known | Novel |
| 10% | MSP | 75.03 | 80.82 | 81.42 | 37.87 | 66.36 | 70.46 | 72.48 | 34.09 |
| | DOC | 74.14 | 79.03 | 79.70 | 30.94 | 72.29 | 74.24 | 76.40 | 35.41 |
| | LMCL | 75.14 | 81.07 | 81.67 | 37.90 | 71.24 | 74.06 | 76.26 | 34.37 |
| | SEG | 68.38 | 73.90 | 74.51 | 30.21 | 64.23 | 66.10 | 67.76 | 36.15 |
| | SEG-SF | 69.67 | 76.17 | 76.78 | 32.74 | 65.73 | 68.31 | 70.16 | 34.90 |
| | GDA | 75.91 | 81.38 | 81.99 | 37.61 | 71.08 | 72.90 | 74.92 | 36.53 |
| | **M3S** | **77.29** | **81.99** | **82.56** | **40.90** | **72.62** | **75.26** | **77.17** | **40.97** |
| 20% | MSP | 74.56 | 80.21 | 80.63 | 53.24 | 66.32 | 70.86 | 72.53 | 44.05 |
| | DOC | 70.83 | 77.42 | 78.06 | 36.65 | 68.82 | 73.29 | 75.54 | 37.42 |
| | LMCL | 75.12 | 80.98 | 81.42 | 53.03 | 67.81 | 73.16 | 75.33 | 38.44 |
| | SEG | 66.19 | 72.31 | 72.88 | 42.57 | 59.72 | 64.18 | 66.33 | 29.63 |
| | SEG-SF | 69.67 | 75.34 | 75.72 | 50.97 | 63.13 | 66.76 | 68.58 | 37.59 |
| | GDA | 75.10 | 80.65 | 81.09 | 52.64 | 67.78 | 72.02 | 73.96 | 40.91 |
| | **M3S** | **75.91** | **81.06** | **81.47** | **55.02** | **69.89** | **73.94** | **75.59** | **47.47** |
| 30% | MSP | 72.00 | 78.17 | 78.54 | 57.63 | 65.73 | 68.74 | 69.74 | 54.75 |
| | DOC | 66.11 | 75.23 | 75.87 | 39.24 | 63.49 | 68.76 | 70.84 | 39.64 |
| | LMCL | 72.06 | 78.72 | **79.09** | 58.25 | 64.18 | 69.44 | 71.09 | 46.36 |
| | SEG | 63.13 | 70.29 | 70.69 | 47.49 | 57.96 | 63.44 | 65.32 | 37.03 |
| | SEG-SF | 68.58 | 75.88 | 76.32 | 51.54 | 62.25 | 66.63 | 68.23 | 44.25 |
| | GDA | 71.77 | 78.07 | 78.46 | 56.57 | 64.08 | 69.13 | 70.98 | 43.20 |
| | **M3S** | **73.23** | **78.77** | **79.09** | **60.70** | **68.12** | **71.18** | **72.18** | **57.17** |

Table 2: Main results (%) with different proportions of OOS relations on FewRel and TACRED datasets. "Known" and "Novel" denote the macro F1 over the known relations and the novel relation, respectively.

classifier in the original SEG method. **GDA** (Xu et al., 2020) utilizes the Mahalanobis distance between each feature and the class prototype as the confidence score.

MSP and GDA require a specific validation set that contains OOS data to adjust their thresholds. We provide such a validation set for MSP and GDA by integrating the validation data of OOS relations. Our method and other baselines only use the data of known relations for training and validation, which is a more realistic setting.

## 6.2 Experimental Settings

We use the HuggingFace's Transformer library[2] to implement the BERT-based model. To ensure a fair comparison, all baselines employ the same BERT encoder. As suggested by Devlin et al. (2019), the learning rate is 1e-5. The temperature scalar $T$ is 0.1. The batch sizes for FewRel and TACRED are 32 and 16. Each self-supervised strategy constructs a batch size pseudo-OOS samples. The hyper-parameters are obtained by a grid search on the validation set.

[2] https://github.com/huggingface

## 6.3 Evaluation Metrics

Following other OOS detection tasks (Yan et al., 2020; Zhang et al., 2021), we use the overall accuracy and the macro F1 score as evaluation metrics. In addition, we report the macro F1 scores of the known relations and the novel relation separately. For each OOS rate, we construct 5 different datasets and report the average results.

## 6.4 Main Results

The main results are shown in Table 2. From these results, we can see that:

(1) Our method M3S significantly outperforms other baselines and achieves state-of-the-art performance on all datasets. For example, compared to GDA, our method achieves 2.38% and 6.56% improvements in terms of the F1 of the novel relation type on FewRel-NRE-20% and TACRED-NRE-20% datasets, respectively. It verifies the effectiveness of our method on the NRE task.

(2) Under each rate, there is a significant performance gap between the baselines and our method. The reason is that previous methods ignore the shallow semantic similarity problem in the NRE task, which makes them difficult to identify OOS data with similar surface information. Besides, we find

| Rate | Method | FewRel | | | | TACRED | | | |
|------|--------|--------|--------|-------|-------|--------|--------|-------|-------|
| | | Accuracy | Macro F1 | Known | Novel | Accuracy | Macro F1 | Known | Novel |
| 20% | **M3S** | **75.91** | **81.06** | **81.47** | **55.02** | **69.89** | **73.94** | **75.59** | **47.47** |
| | - CC | 75.70 | 80.80 | 81.21 | 54.08 | 68.35 | 72.64 | 74.29 | 46.19 |
| | - RR | 75.49 | 80.68 | 81.09 | 54.25 | 68.77 | 72.90 | 74.57 | 46.24 |
| | - IR | 75.31 | 80.47 | 80.89 | 53.98 | 68.83 | 72.94 | 74.62 | 46.18 |
| | - CC & RR | 75.02 | 80.27 | 80.68 | 53.53 | 67.90 | 72.32 | 74.03 | 45.03 |
| | - CC & IR | 74.81 | 80.03 | 80.44 | 53.28 | 67.65 | 71.71 | 73.33 | 45.80 |
| | - RR & IR | 74.72 | 79.97 | 80.38 | 53.41 | 67.50 | 71.55 | 73.26 | 44.17 |
| | - M3S | 69.36 | 77.44 | 78.65 | 0.00 | 64.44 | 68.55 | 72.84 | 0.00 |

Table 3: Ablation studies (%) on the effectiveness of multi-strategy self-supervised learning.

that, without additional OOS validation data, SEG tends to overfit known relations quickly, leading to performance degradation.

(3) Although MSP and GDA achieve acceptable performance, they require additional OOS validation data to adjust their thresholds. Our method does not require such data and still outperforms MSP and GDA methods.

## 6.5 Ablation Study

To gain more insights into the multi-strategy self-supervised learning, we conduct ablation studies by evaluating multiple variants of M3S. The results are shown in Table 3. To reduce the effect of the number of pseudo-OOS data, we keep the total number of pseudo-OOS data per batch for all variants the same as M3S.

From the results, we can see that: Removing any self-supervised strategy, i.e., convex combination (CC), regional replacement (RR), or irregular replacement (IR), brings performance degradation. This proves the effectiveness of each self-supervised strategy. We infer that these strategies can effectively construct diverse pseudo-OOS data to improve performance. When we remove the three strategies (- M3S), the performance drops significantly. This indicates that OOS detection is critical for RE models in real-world applications.

## 6.6 Effect of the Number of Pseudo-OOS Data

We further investigate the effect of varying the number of pseudo-OOS data. As shown in Figure 6, we increase the number of pseudo-OOS data constructed by each self-supervised strategy from 0 to 160. Note that 32 is the default value of our method on the FewRel-NRE dataset. From the results, we can see that:

(1) As shown in Figure 6 (d), with the increase of synthetic OOS data, M3S achieves comparable

performance in terms of the F1 of the novel relation. This proves that our method is robust over a wide range of numbers. Our method is capable of constructing effective and precise pseudo-OOS data for the novel relation type.

(2) For the other metrics, the performance first increases and then slightly drops as the number increases. We speculate that as the number increases, the constructed OOS data grows rapidly, which leads to a severe data imbalance between the novel relation and the known relations. The data imbalance problem makes the model significantly biased towards learning the synthetic OOS data, affecting the performance of known relations. In this paper, we leave this problem for future work.

(3) Compared to the model without pseudo data, our method not only achieves improvements in the novel relation but also brings a positive effect on known relations. This is mainly due to the fact that our method is able to correctly assign the novel relation to OOS data rather than known relations, thus boosting the overall performance.

## 6.7 Efficiency

To demonstrate the efficiency of our method, we compare the average training time per epoch and the total test time of different methods. The results are shown in Figure 7. Even with pseudo-OOS samples, the training time of our method M3S is comparable to that of these baselines. The training time per epoch of our method is even lower than that of the SEG model. Importantly, the test time of our method is comparable to that of the MSP model and significantly lower than other baselines. This is due to the fact that our method does not require additional post-processing modules and it classifies all test data in a uniform manner.

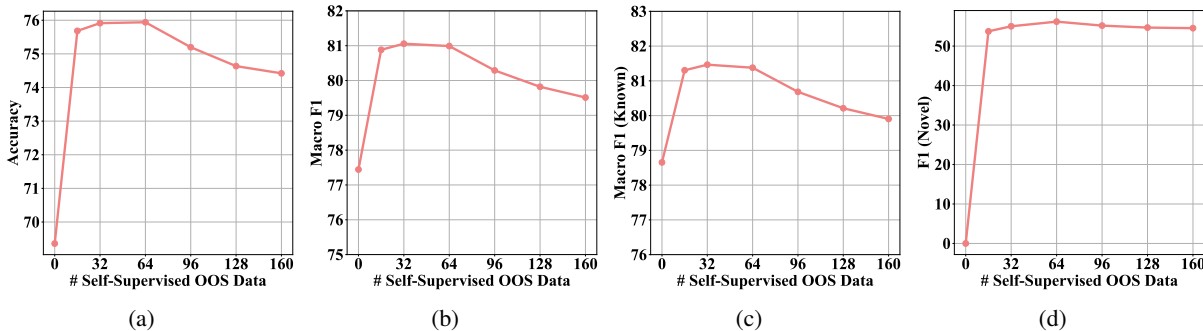

Figure 6: Effect (%) of the number of self-supervised OOS data on the FewRel-NRE-20% dataset. (a), (b), (c), and (d) show the overall accuracy, overall macro F1, macro F1 of known relations, and F1 of the novel relation, respectively. "#" indicates the number of pseudo-OOS data constructed by each self-supervised strategy.

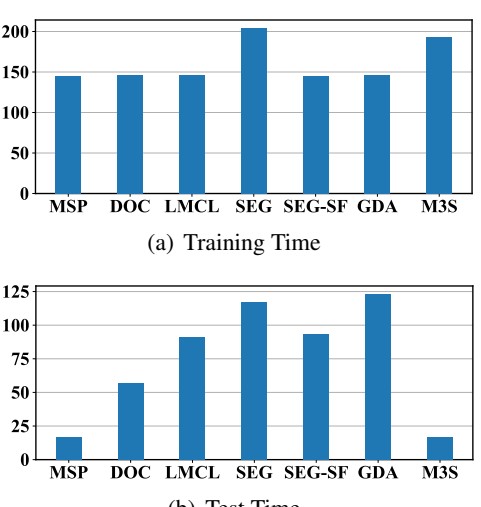

(a) Training Time

(b) Test Time

Figure 7: Comparison (in seconds) of the training time per epoch and the total test time on the FewRel-NRE-20% dataset.

# 7 Related Work

## 7.1 Relation Extraction

Recently, there are many research works on RE (Zelenko et al., 2002; Zeng et al., 2018; Li et al., 2019; Ye et al., 2022). Guo et al. (2019) propose attention-guided graph convolutional networks to select useful substructures from dependency trees for RE. Wang et al. (2021) use a unified label space to model the information between entities and relations. In addition, the few-shot RE task aims to train an RE model using a few samples (Han et al., 2018; Sainz et al., 2021). The continual RE task enables RE models to continually learn new labeled data (Wang et al., 2019; Cui et al., 2021).

Despite the great progress in RE tasks, these existing methods usually ignore the discovery of novel relations, which limits their application in

the real world. Open RE extracts phrases and arguments as specific relations and discovers new relations by clustering or heuristics (Yao et al., 2011; Cui et al., 2018; Kolluru et al., 2022). However, it can not automatically discover novel relations that are not in the predefined relation set. Gao et al. (2019) focus on OOS detection in the few-shot RE task. Compared to these works, we formally propose a realistic and challenging task, i.e., OOS detection for the traditional RE task.

## 7.2 Out-of-Scope Detection

Out-of-scope detection is a long-standing research topic in machine learning, which enables models to identify OOS data as a new/open class (Hodge and Austin, 2004; Zimek et al., 2012; Lee et al., 2018; Zhang et al., 2021). Existing mainstream OOS detection methods can be roughly divided into two categories: probability-based methods (Hendrycks and Gimpel, 2017; Shu et al., 2017) and feature-based methods (Lin and Xu, 2019; Yan et al., 2020; Xu et al., 2020). Probability-based methods derive the confidence score from the probability distribution. Feature-based methods generally employ outlier detection methods, such as LOF (Breunig et al., 2000) or one-class SVM (Schölkopf et al., 2001), to detect OOS data. In addition, there are some research efforts that use synthetic or real OOS data to aid model training (Ryu et al., 2018; Lee et al., 2018; Hendrycks et al., 2019). Zhan et al. (2021) utilize a data augmentation method similar to Mixup (Zhang et al., 2018) to synthesize outliers. Inspired by Zhan et al. (2021) and other data augmentation methods (Yun et al., 2019; Harris et al., 2020), we propose multi-strategy self-supervised learning for the NRE task.

# 8 Conclusion

In this paper, we introduce OOS detection into relation detection, which can automatically discover the data with OOS relations while correctly classifying the data with known relations. To cope with shallow semantic similarity, we propose multi-strategy self-supervised learning. We construct two datasets for the NRE task and compare our method with multiple strong baselines. The results demonstrate the effectiveness of our method.

## Limitations

In this paper, each self-supervised strategy constructs the same amount of pseudo-OOS data. In fact, different strategies bring different improvements as shown in Table 3. Therefore, in future work, we hope to find better ways to fuse these strategies.

## Acknowledgements

This work was supported by the National Natural Science Foundation of China (Grant No.62306087) and the Natural Science Foundation of Shandong Province (Grant No.ZR2023QF154).

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
