# OpenReview forum: "Novel Relation Detection: Discovering Unknown Relation Types via Multi-Strategy Self-Supervised Learning"
_EMNLP/2023/Conference — EMNLP 2023 Findings_

### Official Review · Reviewer_wbgR · 2023-08-03

**Soundness:** 4

**Excitement:**

3: Ambivalent: It has merits (e.g., it reports state-of-the-art results, the idea is nice), but there are key weaknesses (e.g., it describes incremental work), and it can significantly benefit from another round of revision. However, I won't object to accepting it if my co-reviewers champion it.

**Missing References:**

There exists methods on generating OOS/OOD data for this type of problem. It might be worth using these methods as baseline, or discuss the differences. E.g.,
OodGAN: Generative Adversarial Network for Out-of-Domain Data Generation

**Paper Topic And Main Contributions:**

The authors tackle the out-of-scope (OOS) / out-of-distribution (OOD) problem in relation extraction. They try to detect novel (unknown) relation types from known relation types. They first analyze the limitations of existing popular methods including probability-based methods and feature-based methods. They show the confidence distribution and feature space distribution of known and unknown relations are blended and hard to distinguish. They formulate the problem as (k+1) classification and explicitly predict whether an example is of unknown relation type as the k+1 class. Because we do not have training data for this k+1 class, the authors propose different strategies to generate these data from known types. They propose to generate the examples in the format of embeddings of encoder by combining two examples from known types in different ways: convex combination as linear interpolation of values of embeddings of two examples, regional replacement as randomly selecting a span in the embeddings of one example and combining with the rest of the span of the other example, irregular replacement as random selecting dimensions from two examples to construct the new example. By training on these synthetically generated data, the classifier is able to predict this k+1 unknown relation type significantly better than the baseline methods. The authors create the datasets for evaluation from existing relation datasets (FewRel and TACRED) by random sampling the existing relation types as unknown relation types and exclude these types in the training and validation data.


**Reasons To Accept:**

- The authors introduce OOS (OOD) problems in relation extraction, and show limitations of existing methods in terms of confidence and feature space distribution.
- The authors propose a new method of generating OOS/OOD data of unknown type in the feature space (embeddings of encoder) from examples of known types
- The authors compared the method to a few different baseline methods on the created datasets from FewRel and TACRED. They also provide the ablation study on different data generation methods and the amount of generated data. The empirical results are great for the proposed method and also provide insights of the problem for the baseline methods.


**Reasons To Reject:**

- Although the authors compare to quite a few baselines, those methods are limited to confidence-based for output probabilities or distance-based for feature spaces. It would be worth comparing to a baseline that also formulates the problem as k+1 classification and uses synthetic/generated data for training (E.g., using GAN or other heuristics). This would give better estimation on whether the major contribution of the paper (data generation method) is a great or better method in generating OOS/OOD data.
- The authors have a nice visualization on the limitations of probability-based and feature-based methods. It would be interesting to see whether the generated OOD data could be better distinguished in visualization. This would validate the intuition from the authors to mitigate the drawbacks of those methods.


**Reproducibility:**

4: Could mostly reproduce the results, but there may be some variation because of sample variance or minor variations in their interpretation of the protocol or method.

**Reviewer Confidence:**

4: Quite sure. I tried to check the important points carefully. It's unlikely, though conceivable, that I missed something that should affect my ratings.

---

> ### Author Rebuttal · Authors · 2023-08-29
>
> Thank you very much for your constructive comments.
>
> For R1.
>
> A1: Following other OOS detection work, we compare our method with several common OOS detection methods. Comparing our method with some data generation methods (e.g., OodGAN) would indeed enrich this paper. We will investigate feasible data generation methods and add them to the revised version.
>
> For R2.
>
> A2: Additional experiments, such as feature visualization, would indeed enrich this paper. Due to page limitations, we will consider adding them to the revised version.

---

### Official Review · Reviewer_a4GX · 2023-08-04

**Soundness:** 3

**Excitement:**

4: Strong: This paper deepens the understanding of some phenomenon or lowers the barriers to an existing research direction.

**Paper Topic And Main Contributions:**

The paper addresses the challenge of identifying novel relations that are not present in predefined relation ontology and training data. It frames this task as an out-of-scope (OOS) detection problem and proposes a multi-strategy self-supervised learning approach. This approach includes three strategies for constructing pseudo-OOS training instances that simulate novel relations by manipulating the feature representations of known relations. The paper demonstrates improved performance in novel relation classification while maintaining accuracy on known relations.

**Questions For The Authors:**

A: How reliable are the results on TACRED given its small scale of test relations? The construction of the TACRED-NRE dataset includes only the top 20 most frequent relations. Does this imply that all novel relations should be frequent? This assumption might limit the scope of the paper. It would be interesting to see how the proposed model can generalize to unseen long-tail relationships, as novel relations in reality might not occur as frequently as common relations.

**Reasons To Accept:**

1. It is the first to frame the task of novel relation extraction as an OOS detection problem, establishing strong baseline models from OOS detection research. This paper bridges the gap between OOS detection and novel relation extraction.
2. The strategies for constructing pseudo-OOS instances are based on the embedding level, which is efficient and reduces the need for real training data for novel relations. The experimental results demonstrate the effectiveness of these strategies on two datasets (FewRel and TACRED)

**Reasons To Reject:**

1. The assumption that pseudo-OOS instances are novel relations based on arbitrary combinations/corruptions of any two existing relations may not apply to novel relations that don’t satisfy this assumption.
2. The difference between this paper and previous work is not clear. Like previous work, this paper considers novel relations as a whole class, questioning whether the model can distinguish test instances containing novel relations or those without any relations.
3. The paper identifies shallow semantic similarity as an issue in identifying novel relations but does not provide sufficient evidence to demonstrate the severity of this problem or how the proposed model addresses it.

**Reproducibility:**

4: Could mostly reproduce the results, but there may be some variation because of sample variance or minor variations in their interpretation of the protocol or method.

**Reviewer Confidence:**

4: Quite sure. I tried to check the important points carefully. It's unlikely, though conceivable, that I missed something that should affect my ratings.

---

> ### Author Rebuttal · Authors · 2023-08-29
>
> Thank you very much for your constructive comments.
>
> For Q1.
>
> A1: Following other OOS detection work, we adopt a similar data balancing setting. The main purpose of this setting is to allow these datasets to focus on the OOS detection problem rather than the data imbalance problem. This does not mean that the novel relations should be frequent. Our experiments in Table 1 indicate that the "Novel F1" metric of all models decreases when there is less OOS test data, and our model still outperforms other models. The study of unseen relations with long-tailed distributions may indeed become another important research topic in the future work.

---

### Official Review · Reviewer_cVH9 · 2023-08-10

**Soundness:** 3

**Excitement:**

3: Ambivalent: It has merits (e.g., it reports state-of-the-art results, the idea is nice), but there are key weaknesses (e.g., it describes incremental work), and it can significantly benefit from another round of revision. However, I won't object to accepting it if my co-reviewers champion it.

**Paper Topic And Main Contributions:**

In this paper, the authors introduce a new relation extraction task termed "novel relation extraction" (NRE), where the model is required to identify unknown relation types during inference while maintaining its classification capacity for known relation types. The authors highlight the challenge of shallow semantic similarity in the NRE task and demonstrate through experiments that existing probability-based and feature-based Methods might fail under this task. To address the challenge of shallow semantic similarity, this paper enhances the model's robustness to out-of-scope data during the training phase using Multi-Strategy Self-Supervised Learning. This approach achieves state-of-the-art results on two NRE datasets.

**Questions For The Authors:**

1. How do the two hyperparameters, α and β, affect the model? Do they lead to more significant performance improvements when approaching 0?

**Reasons To Accept:**

1. This paper introduces, for the first time, the task of novel relation detection, a task that aligns with the demand for relation extraction models in open-world scenarios.

2. The paper analyzes the limitations of existing mainstream methods and visually demonstrates the reasons for their failures through experimentation.

3. The proposed method presented in this paper is both simple and effective.

**Reasons To Reject:**


1. The pseudo-OOS samples generation methods proposed in this paper lack innovation, as all three methods are essentially variations of the SMOTE [1] technique.

2. The experimental presentation in this paper is insufficient: (1) Lack of case studies to demonstrate the effectiveness of the proposed method in mitigating the impact of shallow semantic similarity; (2) Absence of corresponding experiments to showcase the influence of the proposed method on model feature representation and prediction probabilities, as depicted in Figures 3 and 4.

[1] Chawla N V, Bowyer K W, Hall L O, et al. SMOTE: synthetic minority over-sampling technique[J]. Journal of artificial intelligence research, 2002, 16: 321-357.

**Reproducibility:**

4: Could mostly reproduce the results, but there may be some variation because of sample variance or minor variations in their interpretation of the protocol or method.

**Reviewer Confidence:**

4: Quite sure. I tried to check the important points carefully. It's unlikely, though conceivable, that I missed something that should affect my ratings.

---

> ### Author Rebuttal · Authors · 2023-08-29
>
> Thank you very much for your constructive comments.
>
> For Q1.
>
> A1: These two hyperparameters are indeed very worth exploring. We actually analyzed these two hyperparameters quantitatively during the model design. For example, we have a strategy to limit these two parameters to below 0.2 or even 0.1. However, several strategies we tried all resulted in a slight performance degradation. We speculate that these inductive biases might limit the generated OOS data in some ways so that they cannot match the diverse OOS data in practice.
>
> For R2 (case study, etc.).
>
> A2: Additional experiments, such as case studies and feature visualizations, would indeed enrich this paper. Due to page limitations, we will consider adding them to the revised version.

---

### Official Review · Reviewer_tVzT · 2023-08-11

**Soundness:** 3

**Excitement:**

4: Strong: This paper deepens the understanding of some phenomenon or lowers the barriers to an existing research direction.

**Paper Topic And Main Contributions:**

This paper introduces a new task called novel relation detection (NRD) in relation extraction. Different from traditional RE approaches that recognize predefined relation types only, NRD aims to discover potential new relation types based on training samples of known relations. This paper proposes two NRD datasets and explores various out-of-scope (OOS) detection methods. It further presents an effective NRD method that uses multi-strategy self-supervised learning to address shallow semantic similarity issues in the task. Experimental results demonstrate the effectiveness of the proposed M3S method.

**Questions For The Authors:**

- Since you have run each experiment multiple times, could you conduct a t-test and report the p-values by comparing M3S and the strongest baseline/ablation version in each column in Tables 2 and 3?

- The phenomenon in Figure 6 that the overall accuracy first increases and then drops is a bit strange. I do not quite agree with the authors' speculation because even if about 100 self-supervised OOS samples are generated, this number is still significantly smaller than the number of real training samples per known relation according to Table 1. I wonder if the authors observe the same phenomenon on TACRED.

- Could you provide a more thorough discussion of the limitations of this work?

**Reasons To Accept:**

+ The task of detecting out-of-scope relations makes the relation extraction task more generalizable and has practical value for downstream tasks such as knowledge graph construction.

+ The proposed multi-strategy self-supervised learning is intuitive. Its combination with existing out-of-scope detection methods (i.e., convex combination, regional replacement, and irregular replacement) is well-motivated.

+ Experiments are comprehensive. Two benchmark datasets are used (adapted for the NRD setting). Various out-of-scope detection baselines are compared. The authors also conduct ablation analyses and hyperparameter studies to validate their design choices and configurations.

**Reasons To Reject:**

- The task setting of treating all out-of-scope relations as one class is a bit confusing. In practice, there may be a large number of OOS relations (e.g., 16 in the FewRel dataset used by the authors), and their semantics can be quite scattered in the embedding space. In this case, a more practical setup would be automatically detecting the number of OOS relations (given that the number is in a certain range, e.g., 10-20) through unsupervised clustering.

- Significance tests are missing. It is unclear whether the improvement of M3S over baselines and ablation versions is statistically significant or not. In fact, some gaps in Tables 2 and 3 are quite subtle, therefore p-values should be reported.

- The Limitations section is too concise, different from the clear and detailed writing style in the main body of this paper.

**Reproducibility:**

4: Could mostly reproduce the results, but there may be some variation because of sample variance or minor variations in their interpretation of the protocol or method.

**Reviewer Confidence:**

3: Pretty sure, but there's a chance I missed something. Although I have a good feel for this area in general, I did not carefully check the paper's details, e.g., the math, experimental design, or novelty.

---

> ### Author Rebuttal · Authors · 2023-08-29
>
> Thank you very much for your constructive comments:
>
> For Q1.
>
> A1: We will add significance analysis in the revised version. Due to time limitations, we conducted some of the experiments. For Table 1, we conducted the significance analysis for Rate=20%. The results show that the improvement of our method is statistically significant with p<0.05 under the t-test over the previous strong models, GDA and LMCL. In particular, on the “Novel F1” metric, the improvement of our method is statistically significant with p < 0.01. For Table 2, we conducted a t-test between M3S and -RR. The results show that the improvement of our method is statistically significant with p < 0.05. This can partially prove the significance of the improvement of our method.
>
> For Q2.
>
> A2: It is possible that the description of the coordinates in Figure 6 is not sufficiently clear and caused your misunderstanding. The horizontal coordinate in Figure 6 indicates the number of self-supervised/OOS data constructed by each strategy in each batch, rather than the total number of OOS data constructed during the training process. In this experiment, the data size of known relations in each batch is 32, and the OOS data size of each self-supervised strategy in each batch is shown in the horizontal coordinate. For example, when the horizontal coordinate is 128, the three strategies construct 128*3=384 OOS data per batch, while the data size of known relations in each batch is 32. This will result in a 12-fold difference, leading to a severe data imbalance problem.
>
> For Q3.
>
> A3: Yes, we can. We will optimize the limitation section subsequently. As you suggested above, in terms of task setting, this work treats all novel relations as one class, which might increase the difficulty of artificially formulating the class set in the subsequent labeling process of detected OOS data. In addition, higher proportions (>>30%) of unknown relations may pose more difficulties for all existing models, which needs to be further explored.

---

### Meta-Review · Area_Chair_UYZ1 · 2023-09-19

**Recommendation:** 3

**Metareview:**

The reviews have a consensus on positive evaluations for both soundness and excitement.

Regarding excitement, the paper is the first paper that introduces the task of novel relation detection as an out-of-scope (OOS) detection problem.  The core three methods of pseudo-OOS data generation are largely inspired by prior work (as described in L557-560), and thus the novelty on this aspect is relatively weak.

With respect to soundness, the proposed method is well motivated by problems with existing OOS detection methods and well tested by a comparison with various OOS detection baselines on two benchmark datasets.  The ablation study shows the contribution by each of the three data generation methods.

---

### Decision · Program_Chairs · 2023-10-07

**Decision:**

Accept-Findings

**Comment:**

The reviews have a consensus on positive evaluations for both soundness and excitement.

Regarding excitement, the paper is the first paper that introduces the task of novel relation detection as an out-of-scope (OOS) detection problem.  The core three methods of pseudo-OOS data generation are largely inspired by prior work (as described in L557-560), and thus the novelty on this aspect is relatively weak.

With respect to soundness, the proposed method is well motivated by problems with existing OOS detection methods and well tested by a comparison with various OOS detection baselines on two benchmark datasets.  The ablation study shows the contribution by each of the three data generation methods.